# UNICORNN: UNIMODAL CALIBRATED ORDINAL REGRESSION NEURAL NETWORK

## ABSTRACT

Ordinal regression is a supervised machine learning technique aimed at predicting the value of a discrete dependent variable with an ordered set of possible outcomes. Many of the algorithms that have been developed to address this issue rely on maximum likelihood for training. However, the standard maximum likelihood approach often fails to adequately capture the inherent order of classes, even though it tends to produce well-calibrated probabilities. Alternatively, some methods use Optimal Transport (OT) divergence as their training objective. Unlike maximum likelihood, OT accounts for the ordering of classes; however, in this manuscript, we show that it doesn't always yield well-calibrated probabilities. To overcome these limitations, we introduce UNICORNN, an approach inspired by the well-known Proportional Odds Model, which offers three key guarantees: (i) it ensures unimodal output probabilities, a valuable feature for many real-world applications; (ii) it employs OT loss during training to accurately capture the natural order of classes; (iii) it provides well-calibrated probability estimates through a post-training accuracy-preserving calibration step. Experimental results on six real-world datasets demonstrate that UNICORNN consistently either outperforms or performs as well as recently proposed deep learning approaches for ordinal regression. It excels in both accuracy and probability calibration, while also guaranteeing output unimodality. The code will be publicly available upon acceptance.

## 1 INTRODUCTION

Ordinal regression is an area of supervised machine learning, where the goal is to predict the value of a discrete dependent variable, whose set of (symbolic) possible values is ordered. Despite often being overshadowed by more common tasks like classification and regression, ordinal regression covers a wide range of important applications, such as medical severity grading, credit rating, age estimation, and many more (De Vente et al., 2020; Wienholt et al., 2024; Niu et al., 2016; Kim and Ahn, 2012).

Many practitioners often treat ordinal regression problems as classification or regression problems (for example, this was the case with many submissions to Kaggle's Diabetic Retinopathy competition[1] in 2015). While having common characteristics with both classification and regression, ordinal regression can arguably be viewed as a mid-point between the two. An ordinal model is of course similar to a classification model, in that both predict a discrete value ("label") out of a finite set of possible ones. However, the existence of an order on the set of labels, when available, can potentially lead to improved performance compared to a standard classifier, which does not assume such order exists. This typically occurs via distinguishing between the severity of prediction mistakes: while in classification typically "all mistakes are created equal", in ordinal regression different mistakes may be associated with different severity (for example, in the context of tumor grade prediction, predicting "3" when the ground truth value is "4" may be less severe than a "1" prediction). In regression problems, the dependent variable naturally does take values from an ordered set, however, this set is typically a continuum and is treated numerically, while in ordinal regression it lacks any numerical relation beyond merely order. Therefore, regression performance may be sensitive to monotonic transformations of the dependent variable, while such sensitivity does not take place in ordinal regression problems, as the order is invariant to monotonic transformations. Hence one may

---

[1]https://www.kaggle.com/c/diabetic-retinopathy-detection/

expect that typical ordinal regression algorithms have the potential to outperform classification or regression approaches when the range of the dependent variable is finite and ordered.

A fundamental ordinal regression model is the Proportional Odds Model (POM) (McCullagh, 1980), a generalized linear model similar in spirit to logistic regression, however the logits are defined for their cumulative probabilities. One potential source of sub-optimality of POM and of several recently-proposed approaches for deep ordinal regression, is the often-reasonable requirement that a probabilistic model for ordinal regression will output unimodal probabilities. A $k$-level multinomial distribution is called *unimodal* if there exists $j \in \{1, \ldots, k\}$ such that $\Pr(Y = 1) \leq \ldots \leq \Pr(Y = j) \geq \ldots \geq \Pr(Y = k)$. Although there are domains in which unimodality is not necessarily a desirable property, such as tasks where the most common targets tend to fall at the extremes, in many other real-world domains it is a natural requirement, for example, when predicting a grade of a tumor, it may be counter-intuitive to trust a model prediction which says that a predicted tumor's grade is either "1" or "4", but not "2" or "3". However, unimodality is unfortunately not always fulfilled despite often being a desired characteristic. While this was identified by several recent works for deep ordinal regression (Gao et al., 2017; Diaz and Marathe, 2019; Liu et al., 2019a; 2020), unimodality is often encouraged (but not enforced) via soft targets. However, as we show in Section 2, using soft targets is suboptimal for achieving unimodality.

Another essential feature of an ordinal regression model is its ability to effectively capture the ordered relationship among classes within its training objective, while still reflecting the certainty of the model in its predictions. POM is typically trained via maximum likelihood, similar to several recently proposed deep ordinal regression approaches (Belharbi et al., 2019; Vargas et al., 2020; Fu et al., 2018; Beckham and Pal, 2017; Berg et al., 2020; Vishnu et al., 2019). We argue that maximum likelihood is a sub-optimal measure of quality for ordinal regression setup, as it only considers the probability mass the model assigns to the true class, ignoring the remaining mass. This implicitly assumes that "all mistakes are equal", which, as discussed above, is not the case for ordinal regression. However, a benefit of maximum likelihood is that it tends to yield well-calibrated probabilities. A well-calibrated model ensures that, for example, a 90% predicted probability corresponds to events that actually occur 90% of the time. However, many models, including recent deep ordinal regression approaches, struggle with this. These models often exhibit overconfidence or underconfidence, meaning their probability outputs are unreliable indicators of their true prediction certainty. Another commonly used loss function for ordinal regression tasks is Optimal Transport (OT) divergence. OT excels at capturing the inherent order between labels, potentially making it a better fit. However, as we explain in Section 3.3, OT might lead to peaked output distribution that lacks calibration.

In this manuscript, we therefore focus on two main contributions. First, We present UNICORNN, a novel approach for ordinal regression, based on deep learning machinery, which tackles the three issues pointed out above (i) it contains a mechanism to enforce *unimodality of the output distribution*, implemented via architectural design, (ii) it *effectively captures the ordered relationship among classes* using OT as a training objective (iii) it undergoes a post-training calibration process to output *well-calibrated probability estimates* that reflect the model's confidence in its predictions while still preserving the model's accuracy. Second, as discussed in Section 3.3, we identify a trade-off between certain requirements, noting that OT may prioritize peaked distributions over calibrated ones, which, to the best of our knowledge, was not pointed out in the literature in the context of deep ordinal regression. Importantly, this bonds the requirements together, as other methods that utilize OT as a training objective, end up being uncalibrated. We present experimental results on six real-world image benchmark datasets which demonstrate that UNICORNN consistently performs on par with and often better than several recently proposed approaches for deep ordinal regression in terms of both prediction accuracy and probability calibration while having an unimodality guarantee.

## 2   RELATED WORK

Being a traditional area of machine learning and statistics, there exists a large corpus of literature on ordinal regression. In this section, we focus on approaches based on recent deep-learning architectures. Several such approaches have been proposed in recent years. One common approach is to turn the ordinal regression problem into a multi-label classification problem (Fu et al., 2018; Liu et al., 2017; 2018b; Vishnu et al., 2019; Berg et al., 2020; Cheng et al., 2008; Li et al., 2021). We argue that the multi-label approach has two major problematic aspects: first, the output probabilities are not always

guaranteed to be consistent, in the sense of increasing cumulative distribution (i.e., we would like to predict $\Pr(y \leq 1) \leq \Pr(y \leq 2) \leq \ldots \leq \Pr(y \leq k)$. Second, even if the output probabilities are consistent, as is the case in Liu et al. (2018a); Shi et al. (2023); Cao et al. (2020), the predicted class probabilities are not necessarily unimodal. This is the case in several recent works (Liu et al., 2019b; Vargas et al., 2020; Pan et al., 2018; Kook et al., 2020). Another line of research focuses on addressing ordinal regression by developing unbiased estimator-based approaches that are robust to label noise, as proposed by Garg and Manwani (2020).

One elegant mechanism is to obtain unimodal output probabilities, based on either the Poisson or the Binomial distributions (Beckham and Pal, 2017), which are both unimodal. In both cases, the model outputs a scalar ($\lambda$ in the case of the Poisson, $p$ in the case of the binomial) for each prediction, which is then mapped to a probability mass function that is used (after normalization) as the model output probabilities. Moreover, this method also learns a dataset-wide $\tau$ parameter which controls the shape of the output distribution. While being a convenient architectural-based solution for handling unimodality, this approach is inherently limited in its ability to express the level of uncertainty of the model's prediction. To see why, note that since a single parameter determines both the location of the mode and the decay of the probabilities, the model cannot output a highly flat or highly peaked probability vector, for example.

A different approach to unimodality has been to train the model with soft targets (Gao et al., 2017; Diaz and Marathe, 2019; Liu et al., 2019a; 2020). However, we argue that the utilization of soft targets suffers from two important disadvantages. First, unimodality is only encouraged, but not enforced. In Section 5 we will show cases where models trained with soft targets yield large amounts of non-unimodal predictions. Second, soft targets have a pre-defined decay pattern, which is determined a-priori and hence does not reflect any level of uncertainty with respect to the prediction. Therefore, they are equivalent (in the sense of a 1:1 map) to Dirac predictions (i.e., "one-hot"), and are devoid of any probabilistic insight whatsoever. As we show in this paper, our approach attends to both issues: we guarantee unimodal outputs, by design, and yield well-calibrated probabilities outputs that reflect the model's uncertainty. Other approaches for handling unimodality include Li et al. (2022); Cardoso et al. (2023), where unimodality is encouraged through a dedicated loss term (although not guaranteed), and Belharbi et al. (2019), which employs constrained optimization to achieve unimodality on the training data, but with no guarantees on the predictions on test data.

Several works use cross entropy as a training objective while using one-hot (or binary) targets (Belharbi et al., 2019; Vargas et al., 2020; Fu et al., 2018; Beckham and Pal, 2017; Berg et al., 2020; Vishnu et al., 2019; Cardoso et al., 2023). Also, an accuracy-preserving calibration on a cross-entropy trained model was proposed in (Esaki et al., 2024). However, in the case of one-hot targets, the cross entropy term equals the negative logarithm of the probability assigned by the model to the true class, making it invariant to the distribution of the remaining probability mass. While reasonable in a standard classification setting, this ignores the order of the classes, making it sub-optimal for an ordinal regression setting. Nonetheless, as cross-entropy is a proper-scoring rule, it tends to yield calibrated probability estimates (Lakshminarayanan et al., 2017). To address cross-entropy's limitation in handling the order of classes, some approaches use OT loss (Hou et al., 2016; Beckham and Pal, 2017; Liu et al., 2019a), which is a natural way to incorporate the order of the classes into the loss term. However, OT tends to favor peaked output distributions instead of calibrated ones (see Section 3.3). UNICORNN aims to benefit from both the ability to capture the order of classes using OT as a training objective, and the generation of calibrated probability estimates through accuracy-preserving calibration.

In summary, to the best of our knowledge, no existing work has successfully met all three fundamental requirements for an effective ordinal regression model: (i) Ensuring unimodality in the output distribution, ideally through architectural design; (ii) Aligning the training objective function with the ordinal nature of the label space; (iii) Reflecting the model's uncertainty in the decay of the output probabilities, preferably with well-calibrated probabilities. These requirements led to the development of UNICORNN, presented in this paper.

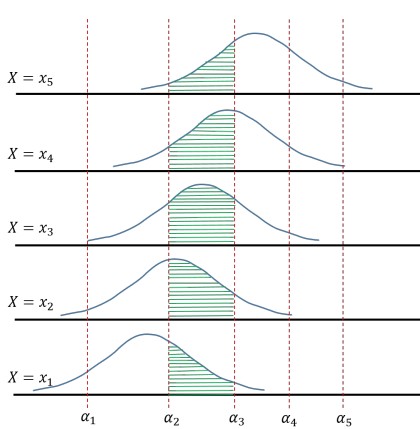

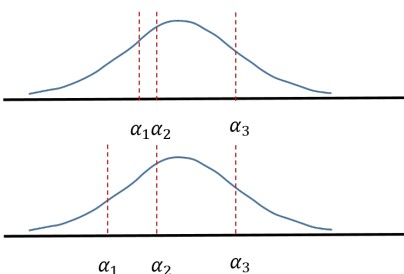

Figure 2: POM deficiencies. The plots illustrate two instances of POM, which highlight its two deficiencies: (1) POM does not always output unimodal probabilities: the first plot shows an example where the output probabilities are such that $\Pr(Y = 1) > \Pr(Y = 2) < \Pr(Y = 3) > \Pr(Y = 4)$. (2) The likelihood function of POM is invariant to the way the predicted probability mass of the incorrect classes is assigned: if the correct class is 3, both instances have the same likelihood, even though in the second instance, the probability mass assigned to neighboring class 2 is larger.

Figure 1: The proportional odds model. $x_i$ is a realization of $X$. The standard logistic density is shifted by $\beta^T x_i$. The thresholds $\alpha_j$ define the bins which determine the probability predicted by the model to each class. For example, the green area defines the probability $\Pr(Y = 3)$ for given $\alpha_1, \ldots, \alpha_5$ and $\beta$.

# 3 PRELIMINARIES

## 3.1 THE PROPORTIONAL ODDS MODEL

Let $(X, Y) \in \mathcal{X} \times \mathcal{Y}$ be random variables, having joint probability $\mathcal{P}_{XY}$, where $\mathcal{X} = \mathbb{R}^d$, $\mathcal{Y} = \{1, \ldots, k\}$, and $1, \ldots, k$ are considered as symbols. Let $\preceq$ be an order relation defined on $\mathcal{Y}$ such that $1 \preceq \ldots \preceq k$. The proportional odds model is parametrized by $\alpha \in \mathbb{R}^{|\mathcal{Y}|-1}$, $\beta \in \mathbb{R}^d$ and applies to data $\{(x_i, y_i)\}_{i=1}^n$, sampled i.i.d from $\mathcal{P}_{XY}$. Let $\epsilon$ be a logistic random variable (thus having a sigmoid cumulative distribution function $F(x) = \frac{1}{1+\exp(-x)}$), and let $\mathcal{Z}$ be a random variable defined as $\mathcal{Z} = \beta^T \mathcal{X} + \epsilon$. The entries of $\alpha$ are used to define the cumulative conditional probabilities via

$$\mathbb{P}(Y \preceq j | X = x) = \Pr(Z \leq \alpha_j) = F(\alpha_j - \beta^T x) \tag{1}$$

Similarly to logistic regression, this yields linear log-odds (logits), however, defined with respect to cumulative terms

$$\gamma_j \equiv \log \frac{\mathbb{P}(Y \preceq j | X = x)}{\mathbb{P}(Y \succ j | X = x)} = \alpha_j - \beta^T x$$

It is convenient to interpret equation 1 by viewing $\beta^T x$ as a factor that shifts the standard logistic density function, while the $\alpha_j$ terms are thresholds, with respect to which the cumulative probabilities are defined. This is depicted in Figure 1.

Let $(x, y)$ be a realization of $(X, Y)$. The likelihood assigned by the model to $(x, y)$ is

$$L(\alpha, \beta; (x, y)) = \mathbb{P}(Y = y | X = x; \alpha, \beta) = F(\alpha_y - \beta^T x) - F(\alpha_{y-1} - \beta^T x), \tag{2}$$

considering $\alpha_0 = -\infty$ and $\alpha_k = \infty$. The model is typically trained in a standard fashion by maximizing the log-likelihood function on the training data.

Despite its popularity, POM has two key limitations: First, the model's output probabilities are not necessarily unimodal (see Figure 2). Second, the likelihood function in equation 2 depends only on the probability the model assigns to the correct class $y$ and is invariant to the way the remaining probability mass is assigned by the model. This ignores the order on the label set, and hence does not use important information that might be used to improve prediction quality, as depicted in Figure 2. This is also true of ordinal likelihood proposed in Chu et al. (2005). In addition, it is important to

mention that as the cross-entropy term is essentially equivalent to the model's negative log-likelihood function, this invariance to the partition of the remaining mass over the incorrect classes is common to all models trained via cross-entropy minimization, as long as the target labels are one-hot. In Section 4 we will show how UNICORNN overcomes these two limitations of POM.

## 3.2 PROBABILITY CALIBRATION

Probability Calibration refers to the alignment between the predicted probabilities generated by a classification model and the true, empirical probabilities observed in the data. For instance, it is anticipated that when a classification model assigns a probability of 0.8 to class $i$ for a certain sample, approximately $80\%$ of those samples would indeed belong to class $i$ based on their ground truth labels. Formally, the concept of calibration can be defined as:

$$\mathbb{P}[y = i | p_i(x) = q] = q, \quad \forall q \in [0, 1], i \in [k] \tag{3}$$

Here, $k$ denotes the number of classes, $x$ and $y$ represent the model input and the ground truth class, respectively, and $p_i(x)$ signifies the model's output probability for class $i$ given input $x$. The probability is taken over the joint distribution of $x, y$.

Alternatively, if we denote $y$ as a one-hot vector indicating the ground truth label, we can express an equivalent definition to equation 3: A classification model is calibrated when $\mathbb{E}_{x,y}[y|p(x)] = p(x)$. In practice, $p(x)$ is a one-to-one function, hence we can refine the definition further: $\mathbb{E}_{x,y}[y|x] = p(x)$.

To achieve calibration, minimizing the squared $L_2$ norm on the difference between the two terms is desirable. However, given the unknown *real* joint distribution $x, y$, we can only approximate $\mathbb{E}_{x,y}[y|x]$. One approach is to use a Monte Carlo approximation with one sample, yielding the approximation $y$, i.e., minimize the following:

$$BS(y, p(x)) = \|y - p(x)\|_2^2 \tag{4}$$

This term is known as the Brier Score (BS) (Brier, 1950). BS is a well-known Proper Scoring Rule (Dawid and Musio, 2014), which evaluates the accuracy of probabilistic predictions. Proper scoring rules are maximized when the probabilistic forecast matches the true probability distribution of outcomes. Therefore, we will use BS as a training objective to calibrate the model predictions.

**Accuracy-Preserving Calibration** An accuracy-preserving calibration method is a technique that adjusts the probability outputs of a pre-trained model to improve their calibration, without affecting the model's accuracy. A popular method is Temperature Scaling (TS) (Guo et al., 2017), which learns a single temperature parameter that is used to rescale the model's logits before applying the softmax activation, adjusting the confidence predictions. Adaptive Temperature Scaling (ATS) (Balanya et al., 2024) extends TS such that instead of a single temperature parameter, it learns a mapping $x \mapsto T(x)$ that adaptively scales the logits based on the input $x$. As part of UNICORNN we define an accuracy-preserving calibration method with similarity to TS and ATS which is trained via Brier Score (Section 4.3).

## 3.3 OPTIMAL TRANSPORT

Let M be a finite metric space with moving cost metric $c(x, y)$ between elements $x, y \in M$, and let $p, q$ be probability mass functions on $M$. The optimal transport (OT) or 1-Wasserstein distance between $q$ and $p$ is defined as:

$$OT(p, q) = \inf_{\gamma \in \Gamma} \int_{M \times M} c(x, y) d\gamma(x, y), \tag{5}$$

Where $\Gamma$ is the set of joint probabilities on $M \times M$ with marginals $q$ and $p$, and $c$ specifies moving costs between elements of $M$. This computes the optimal way to transport $q$ into $p$. When $q$ is a Dirac (one-hot), OT simplifies to:

$$OT(p, q) = \sum_{i=1}^{k} p_i c(i, j), \tag{6}$$

Where $j$ is the correct class and $k$ is the number of classes. With model outputs $p$ and one-hot target $q$, this loss is differentiable w.r.t $p$. The cost metric $c$ can encode domain knowledge. For ordered classes $M = 1, ..., k$, a natural cost is $c(i, j) = |i - j|^m$ for some $m \geq 1$.

Importantly, one drawback of OT as a loss function, is its tendency to prioritize peaked distributions over the actual probabilities. For instance, Let $x$ represent a sample data point, $y$ denote the class label associated with $x$, and $q$ be the one-hot encoded vector representation of $y$. Consider the conditional probabilities $\mathbb{P}[y = 1|x] = 0.25, \mathbb{P}[y = 2|x] = 0.5, \mathbb{P}[y = 3|x] = 0.25$. Now, suppose there are two model outputs: $p_1 = [0.25, 0.5, 0.25]$ and $p_2 = [0, 1, 0]$. To achieve calibration, the model should ideally output $p_1$ as it equals the conditional probability distribution $y|x$. However, it is notable that for the cost metric $c(i, j) = |i - j|$, we find that $OT(p_1, q) = 1$ with a combined probability of $0.25 + 0.25 = 0.5$, and $OT(p_1, q) = 0.5$ with a probability of $0.5$. Consequently, $\mathbb{E}_{y|x}[OT(p_1, q)] = 0.75$, and similarly, $\mathbb{E}_{y|x}[OT(p_2, q)] = 0.5$. This observation indicates that OT tends to favor peaked distributions over calibrated ones. UNICORNN addresses this challenge, as described in Section 4.3.

# 4 UNICORNN

In this section we describe our novel mechanism for UNImodal Calibrated Ordinal Regression Neural Network, UNICORNN an approach for architectural-based generation of unimodal output probability distributions, as well as accuracy-preserving probability calibration.

## 4.1 RATIONAL

Achieving unimodality directly via architectural design has a major advantage since the output probabilities are guaranteed to be unimodal for every input instance, as is also the case for the mechanism proposed in Beckham and Pal (2017). However, UNICORNN employs the truncated normal distribution, depending on two parameters $(\mu, \sigma)$, which influence the location of the mode and the decay of the probability mass. This adds flexibility to the shape of the output probability vector, compared to the mechanism in Beckham and Pal (2017) where a single parameter determines both the mode and the decay. For determining the mode of the distribution, a map $x \mapsto \mu$ is learned via OT using equation 6. However, as mentioned in Section 3.3, OT tends to favor peaked distributions over calibrated ones. To address this issue, we adopt an accuracy-preserving calibration strategy where, subsequent to learning a map $x \mapsto \mu$, the model further learns a map $x \mapsto \sigma$ by optimizing the Brier Score using equation 4, while maintaining the previously learned map $x \mapsto \mu$ fixed.

## 4.2 UNIMODAL OUTPUT PROBABILITIES GENERATION

Inspired by POM, we utilize thresholds to define bins, so that the total mass inside each bin is the output probability of the corresponding class. However, we observe that the lack of unimodality of POM can be fixed by letting the bins be of equal length and remain fixed during training.

Therefore, instead of learning the thresholds, during training two maps $x \mapsto \mu, x \mapsto \sigma$ are learned, where $\mu$ is a location parameter, and $\sigma$ is a scale parameter. Both define a truncated normal distribution $\mathcal{N}_{\text{trunc}}(\mu, \sigma^2, -1, 1)$ (a normal distribution which having the same density as the normal density on [-1,1], normalized to have a unit integral, and is zero outside this interval), from which the output probabilities are derived.

Formally, we divide the range $[-1, 1]$ to $k$ equal bins similarly to da Costa et al. (2008), where $k$ is the number of classes, defined by $-1 = \alpha_0, \alpha_1, ..., \alpha_k = 1$, so that $\alpha_i - \alpha_{i-1} = \frac{2}{k}$. The probabilities are given by

$$p_i(x) = \mathbb{P}(Y = i|X = x) = F_{\mu(x),\sigma(x)}(\alpha_i) - F_{\mu(x),\sigma(x)}(\alpha_{i-1}), \tag{7}$$

where $F_{\mu,\sigma}(\cdot)$ is the $\mathcal{N}_{\text{trunc}}(\mu, \sigma^2, -1, 1)$ cumulative distribution function, and note that $\mu, \sigma$ are in fact functions of the input instance $x$.

To compensate for the fact that the probability-generating mechanism depends on fewer parameters than POM (2 for the former, $d + k - 1$ for the latter), the maps $x \mapsto \mu, x \mapsto \sigma$ are expressed via two deep neural networks (DNNs) which share a common backbone model, hence can represent a complex

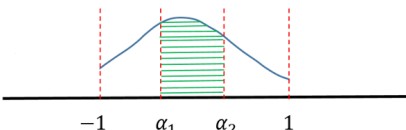

Figure 3: Generation of unimodal output probabilities for $k = 3$ classes. An input $x$ is mapped to a $(\mu, \sigma)$ pair, which defines a truncated normal distribution $\mathcal{N}_{\text{trunc}}(\mu, \sigma^2, -1, 1)$ over the real line. The output probabilities are proportional to the mass in the bins, which are of equal length. The green area equals to probability $p_2(x)$, corresponding to $\mathbb{P}(Y = 2 | X = x)$.

nonlinear relation. Our proposed mechanism for the generation of unimodal output probabilities is depicted in Figure 3.

The following lemma, proved in Appendix A , establishes that the model output probabilities are indeed unimodal.

**Lemma 1.** *Let $x \in \mathbb{R}^d$ be an input to the model, which is mapped to $\mu = \mu(x), \sigma = \sigma(x)$. Let $p_1, \ldots p_k$ be the model output probabilities, generated via equation 7. Then $p_1, \ldots p_k$ define a unimodal multinomial random variable.*

### 4.3 Accuracy-Preserving Calibration in UNICORNN

UNICORNN introduces a re-training procedure for $x \mapsto \sigma$ mapping while preserving the already trained mapping $x \mapsto \mu$. The learning of $x \mapsto \sigma$ is done by minimizing the Brier Score loss in equation 4, which ensures well-calibrated probabilities and preserves the model accuracy. Since the mapping $x \mapsto \sigma$ controls the decay of the generated probabilities and the mapping $x \mapsto \mu$ which controls the mode remains unchanged, the accuracy is not affected by the calibration, resulting in an accuracy-preserving calibration. We state it in the following lemma (proved in Appendix B).

**Lemma 2.** *Let $x \in \mathbb{R}^d$ be an input to the model, mapped to $\mu = \mu(x)$, $\sigma_1 = \sigma_1(x)$. Let $\sigma_2 = \sigma_2(x)$ be a re-trained mapping via minimization of loss using equation 4. Let $p_1^{\sigma_1}, \ldots, p_k^{\sigma_1}$ and $p_1^{\sigma_2}, \ldots, p_k^{\sigma_2}$ be the model output probabilities generated using $(\mu, \sigma_1)$ and $(\mu, \sigma_2)$, respectively, via equation 7. Then:*

$$\text{argmax}_{1 \le i \le k} \, p_i^{\sigma_1} = \text{argmax}_{1 \le i \le k} \, p_i^{\sigma_2}.$$

### 4.4 Training Procedure

To summarize, UNICORNN's training process consists of two distinct phases, as detailed in Algorithms 1, 2. In the first phase, described in Algorithm 1, the parameters $\theta_\mu$ and $\theta_\sigma$ of $\mu(\cdot)$ and $\sigma(\cdot)$, respectively, along with the backbone parameters $\phi$, are jointly optimized using Gradient Descent to minimize the OT loss defined in Eq. 6.

In the second phase, outlined in Algorithm 2, the parameters $\theta_\mu$ and $\phi$ are kept fixed, while $\theta_\sigma$ is further optimized via Gradient Descent to minimize the Brier Score (BS) defined in Eq. 4.

---

**Algorithm 1** Optimal Transport-based Training

---

**Input:** Training data $\mathcal{D} = \{(x_i, y_i)\}_{i=1}^N$, number of classes $k$, batch size $b$, number of epochs $T$
**Output:** Parameters $\theta_\mu, \theta_\sigma$ of $\mu(\cdot)$ and $\sigma(\cdot)$, respectively, and backbone parameters $\phi$
  1: Initialize parameters $\theta_\mu, \theta_\sigma, \phi$
  2: **for** epoch $t = 1$ to $T$ **do**
  3:     **for** each batch $\mathcal{B} \subseteq \mathcal{D}$ of size $b$ **do**
  4:         Compute $\mu_{\phi, \theta_\mu}(x_i)$ and $\sigma_{\phi, \theta_\sigma}(x_i)$ for all $x_i \in \mathcal{B}$ and calculate $p(x_i)$ using Eq. 7
  5:         Compute the loss $\mathcal{L}_{\text{OT}}(\mathcal{B}) = \frac{1}{b} \sum_{(x_i, y_i) \in \mathcal{B}} \text{OT}(p(x_i), y_i)$ using Eq. 6
  6:         Update $\theta_\mu, \theta_\sigma, \phi$ via Gradient Descent
  7:     **end for**
  8: **end for**
  9: **return** $\theta_\mu, \theta_\sigma, \phi$

---

---

**Algorithm 2** Accuracy Preserving Calibration

---

**Input:** Training data $\mathcal{D} = \{(x_i, y_i)\}_{i=1}^N$, number of classes $k$, batch size $b$, number of epochs $T$, parameters $\theta_\mu$ of $\mu(\cdot)$ and backbone parameters $\phi$

**Output:** Parameters $\theta_\sigma$ of $\sigma(\cdot)$

1: **for** epoch $t = 1$ to $T$ **do**
2:    **for** each batch $\mathcal{B} \subseteq \mathcal{D}$ of size $b$ **do**
3:       Compute $\mu_{\phi,\theta_\mu}(x_i)$ and $\sigma_{\phi,\theta_\sigma}(x_i)$ for all $x_i \in \mathcal{B}$ and calculate $p(x_i)$ using Eq. 7
4:       Compute the loss $\mathcal{L}_{\mathrm{BS}}(\mathcal{B}) = \frac{1}{b} \sum_{(x_i, y_i) \in \mathcal{B}} \mathrm{BS}(p(x_i), y_i)$ using Eq. 4
5:       Update **only** $\theta_\sigma$ via Gradient Descent
6:    **end for**
7: **end for**
8: **return** $\theta_\sigma$

---

## 5 EXPERIMENTAL RESULTS

### 5.1 DATASETS

We evaluate UNICORNN on six real-world benchmark image datasets, involving various ordinal regression tasks: age-detection (Adience Eidinger et al. (2014), FG-Net Fu et al. (2014), AAF Cheng et al. (2019)), facial beauty prediction (SCUT-FBP5500 Liang et al. (2018)), bio-medical image classification (Retina-MNIST Yang et al. (2021)) and image aesthetics estimation (EVA Kang et al. (2020)). A more detailed description of the datasets appears in Appendix D . Some examples from the Adience and Retina-MNIST datasets are shown in Figure 4.

### 5.2 BENCHMARK

We compare UNICORNN to five recently presented approaches for deep ordinal regression, with unimodal output probabilities and to a deep learning approach of POM:

**DLDL (Gao et al., 2017)**, an approach utilizing soft labels, generated using squared exponentially decaying distributions, trained using Kullback-Leibler divergence minimization (equivalent to cross-entropy minimization).

**SORD (Diaz and Marathe, 2019)**, an approach utilizing soft labels, generated using linear exponentially decaying distributions, trained using Kullback-Leibler divergence minimization.

**Beckham and Pal (2017)**, an architectural-based approach in which unimodal output probabilities are generated using the binomial distribution (single-learned parameter), trained using optimal transport loss.

**Liu et al. (2019a)**, an approach utilizing soft labels, created as a mixture of Dirac, uniform, and linear exponentially decaying distributions, trained using optimal transport loss.

**POM (McCullagh, 1980)**, a variant of the POM, incorporating a deep learning model with a POM layer integrated on top, trained using cross-entropy loss.

**UnimodalNet (Cardoso et al., 2023)**, a non-parametric architectural-based approach for unimodality, trained using cross-entropy loss.

To perform a fair comparison, we implemented all methods, using the same image transformations, backbone CNN and training procedures, so that the methods differ only in their output layer architectures and loss functions. We performed 5 independent trials, using the same train-validation-test splits for all methods. Additional technical details can be found in Appendix E. For reproducibility, the supplementary material contains code reproducing the results reported in this section.

### 5.3 EVALUATION METRICS

We report several commonly-used evaluation metrics for ordinal regression tasks: Mean Absolute Error (MAE), One-Off Accuracy (OOA), Spearman correlation, Quadratic Weighted Kappa (QWK), as well as the percentage of test examples with unimodal predicted output probabilities.

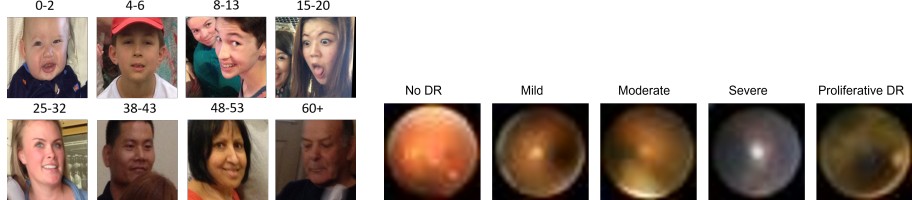

Figure 4: Left: Examples from the Adience dataset. The age category is indicated above each image. Right: Examples from the Retina mnist dataset. Diabetic Retinopathy classes are indicated above.

In addition, following the discussion in section 3.2, we evaluate the model's probability calibration using the well known Expected Calibration Error (ECE). ECE quantifies the discrepancy between predicted probabilities and actual outcomes, measuring how well the predicted probabilities of a model align with the true likelihood of the predicted events. To compute ECE, the $[0, 1]$ interval is first divided into a set of $b$ equal-length bins. ECE is defined as:

$$ECE = \sum_{i=1}^{b} \frac{|B_i|}{|X|} |\text{Acc}(B_i) - \text{Conf}(B_i)|,$$

where $b$ is the number of probability bins (in all of our experiments $b = 10$), $\text{Acc}(B_i)$ is the empirical accuracy in bin $B_i$ and $\text{Conf}(B_i)$ is the average predicted probability in bin $B_i$. An ECE of 0 indicates perfect calibration, while higher values signify miscalibration, with the model being over-confident or under-confident.

### 5.4 RESULTS ON REAL WORLD DATASETS

Table 1 shows the test results of each method on the six benchmark datasets. As can be seen, UNICORNN performs at least on par and often better than the compared baselines, in a fairly consistent manner, across the various datasets and evaluation metrics, specifically with respect to the MAE, where UNICORNN outperforms on all the six datasets. In addition, observe that only UNICORNN, UnimodalNet and Beckham and Pal (2017) output unimodal probabilities, via architectural design, while the other baselines, trained using soft targets, do not always output unimodal probabilities. Yet, unlike UnimodalNet and Beckham and Pal (2017) which tend to produce poorly calibrated probability estimates, UNICORNN outperforms all baselines in terms of probability calibration across five datasets, while on the EVA dataset, the difference from SORD is negligible.

### 5.5 ABLATION STUDY

In this section, we analyzed the impact of the calibration phase introduced in Section 4.3 on the ECE values of UNICORNN. Table 2 shows the ECE with and without the calibration phase on the Adience, EVA, AAF, and Retina MNIST datasets. Without calibration, the ECE is higher, indicating that the model's outputs are not well-calibrated due to the OT properties discussed in Section 3.3. This highlights the importance of incorporating the calibration step to accurately reflect the model's confidence in its predictions.

## 6 CONCLUSION

In this manuscript, we identify several issues with current deep ordinal regression methods, including the potential for OT to result in poor probability calibration. We therefore presented UNICORNN, an approach for deep ordinal regression, inspired by the proportional odds model. UNICORNN utilizes an architectural mechanism for the generation of unimodal output probabilities, trained using OT objective and calibrated using an accuracy-preserving calibration process which encourages uncertainty awareness. We demonstrated that while performing on par with and often better than other recently proposed approaches for ordinal regression, the presented method enjoys the benefits of *guaranteed* unimodal and well-calibrated output probabilities.

Table 1: Performance of various methods on real world datasets, in a mean $\pm$ std format

| Dataset | Method | MAE↓ | OOA↑ | Spearman↑ | QWK↑ | % Unimodal↑ | ECE ($b = 10$)↓ |
|---|---|---|---|---|---|---|---|
| Adience | Beckham and Pal | .5 ± .06 | .93 ± .01 | **.9 ± .01** | **.91 ± .02** | **1 ± 0** | .23 ± .05 |
| | Liu et al. | .47 ± .05 | .94 ± .01 | .89 ± .02 | .9 ± .02 | .47 ± .04 | .21 ± .03 |
| | DLDL | .49 ± .06 | .93 ± .01 | .88 ± .02 | .89 ± .03 | .62 ± .08 | .42 ± .04 |
| | SORD | .47 ± .06 | .94 ± .01 | .89 ± .02 | .9 ± .02 | .99 ± .003 | .14 ± .03 |
| | POM | .48 ± .06 | .94 ± .01 | .89 ± .03 | **.91 ± .03** | .81 ± .04 | .32 ± .04 |
| | UnimodalNet | .49 ± .06 | .93 ± .01 | .88 ± .02 | .90 ± .03 | **1 ± 0** | .34 ± .04 |
| | UNICORNN | **.46 ± .05** | **.95 ± .01** | .9 ± .02 | .91 ± .02 | **1 ± 0** | **.07 ± .03** |
| Retina MNIST | Beckham and Pal | .8 ± .02 | .79 ± .01 | .58 ± .02 | .55 ± .02 | **1 ± 0** | .17 ± .01 |
| | Liu et al. | .68 ± .02 | **.82 ± .01** | **.61 ± .02** | .58 ± .02 | .72 ± .03 | .27 ± .01 |
| | DLDL | .72 ± .02 | .81 ± .01 | .59 ± .02 | .58 ± .01 | .98 ± .01 | .29 ± .01 |
| | SORD | .75 ± .02 | .78 ± .01 | .58 ± .007 | .57 ± .007 | .87 ± .03 | .11 ± .01 |
| | POM | .83 ± .02 | .76 ± .01 | .53 ± .02 | .53 ± .02 | .43 ± .03 | .17 ± .01 |
| | UnimodalNet | .74 ± .02 | .8 ± .01 | .59 ± .01 | .58 ± .01 | **1 ± 0** | .14 ± .01 |
| | UNICORNN | **.67 ± .009** | **.82 ± .007** | **.61 ± .01** | **.59 ± .01** | **1 ± 0** | **.06 ± .01** |
| FG-NET | Beckham and Pal | .44 ± .06 | .95 ± .03 | .80 ± .04 | .79 ± .03 | **1 ± 0** | .16 ± .02 |
| | Liu et al. | .32 ± .08 | .96 ± .02 | .86 ± .03 | .84 ± .04 | .01 ± .01 | .08 ± .02 |
| | DLDL | .4 ± .09 | .96 ± .02 | .81 ± .03 | .81 ± .03 | .12 ± .04 | .49 ± .06 |
| | SORD | .36 ± .08 | .97 ± .02 | .84 ± .05 | .82 ± .04 | .99 ± .01 | .26 ± .05 |
| | POM | .33 ± .06 | .97 ± .02 | .84 ± .04 | .85 ± .04 | .41 ± .06 | .21 ± .03 |
| | UnimodalNet | .36 ± .07 | .97 ± .01 | .84 ± .02 | .84 ± .03 | **1 ± 0** | .25 ± .04 |
| | UNICORNN | **.31 ± .08** | **.99 ± .02** | **.87 ± .03** | **.88 ± .01** | **1 ± 0** | **.07 ± .02** |
| AAF | Beckham and Pal | .61 ± .13 | .91 ± .05 | .81 ± .03 | .8 ± .05 | **1 ± 0** | .16 ± .04 |
| | Liu et al. | .43 ± .01 | **.97 ± .005** | .82 ± .01 | **.85 ± .01** | .7 ± .17 | .22 ± .01 |
| | DLDL | .54 ± .02 | .95 ± .01 | .77 ± .01 | .77 ± .02 | .95 ± .02 | .32 ± .02 |
| | SORD | .44 ± .02 | .96 ± .01 | .82 ± .01 | .84 ± .01 | **1 ± 0** | .14 ± .02 |
| | POM | .45 ± .01 | .96 ± .01 | .80 ± .01 | .83 ± .01 | .48 ± .07 | .28 ± .02 |
| | UnimodalNet | .45 ± .01 | .96 ± .01 | .80 ± .01 | .83 ± .01 | **1 ± 0** | .30 ± .01 |
| | UNICORNN | **.42 ± .01** | **.97 ± .005** | **.83 ± .01** | **.85 ± .005** | **1 ± 0** | **.03 ± .01** |
| SCUT-FBP5500 | Beckham and Pal | .62 ± .12 | .92 ± .04 | .84 ± .02 | .81 ± .05 | **1 ± 0** | .19 ± .08 |
| | Liu et al. | .54 ± .03 | .95 ± .01 | .83 ± .01 | .82 ± .01 | .18 ± .14 | .27 ± .02 |
| | DLDL | .68 ± .05 | .9 ± .01 | .79 ± .02 | .73 ± .02 | .86 ± .08 | .28 ± .04 |
| | SORD | .57 ± .03 | .94 ± .01 | .83 ± .01 | .81 ± .01 | .99 ± .005 | .09 ± .02 |
| | POM | .53 ± .03 | .95 ± .01 | .83 ± .01 | .83 ± .01 | .28 ± .04 | .33 ± .03 |
| | UnimodalNet | .52 ± .02 | .95 ± .01 | .82 ± .01 | .83 ± .01 | **1 ± 0** | .35 ± .01 |
| | UNICORNN | **.47 ± .02** | **.96 ± .01** | **.85 ± .01** | **.85 ± .01** | **1 ± 0** | **.06 ± .02** |
| EVA | Beckham and Pal | .61 ± .03 | .93 ± .01 | **.6 ± .01** | **.6 ± .02** | **1 ± 0** | .11 ± .01 |
| | Liu et al. | .65 ± .02 | .91 ± .02 | .53 ± .01 | .52 ± .02 | .72 ± .05 | .31 ± .01 |
| | DLDL | .66 ± .03 | .91 ± .01 | .53 ± .02 | .52 ± .02 | .98 ± .01 | .2 ± .01 |
| | SORD | .6 ± .02 | .93 ± .01 | .58 ± .02 | .57 ± .02 | **1 ± 0** | **.07 ± .02** |
| | POM | .65 ± .03 | .91 ± .01 | .53 ± .02 | .53 ± .03 | .89 ± .02 | .45 ± .02 |
| | UnimodalNet | .66 ± .03 | .91 ± .01 | .52 ± .02 | .52 ± .02 | **1 ± 0** | .45 ± .02 |
| | UNICORNN | **.57 ± .01** | **.94 ± .005** | **.6 ± .01** | .58 ± .01 | **1 ± 0** | .08 ± .02 |

Table 2: The effect of UNICORNN's calibration phase on the ECE for the Adience and Retina MNIST datasets.

| Dataset | Post-hoc Calibration | ECE ($b = 10$)↓ |
|---|---|---|
| Retina MNIST | yes | **.06 ± .01** |
| | no | .38 ± .007 |
| Adience | yes | **.07 ± .03** |
| | no | .18 ± .03 |
| AAF | yes | **.03 ± .01** |
| | no | .28 ± .01 |
| EVA | yes | **.08 ± .02** |
| | no | .32 ± .02 |

**Reproducibility Statement** All technical details of the experiments are provided in Section E, also the code will be made publicly available upon acceptance of this work.

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

## A    PROOF OF LEMMA 1

*Proof.* Let $p_i, p_{i+1}$ be the output probabilities of two adjacent classes, and let $-1 = \alpha_0, \ldots, \alpha_k = 1$ be the thresholds. We will show that (i) if $\mu \leq \alpha_{i-1}$ then $p_i \geq p_{i+1}$. Symmetrical argument will then imply that if $\mu \geq \alpha_{i+1}$ then $p_{i+1} \geq p_i$ (ii) if $\mu \in (\alpha_{i-1}, \alpha_i)$ then $p_i > p_{i+1}$, whenever the latter exists. Similarly, this would imply that $p_i > p_{i-1}$. Together, (i) and (ii) will prove the statement of the lemma.

Denote by $f$ the density of the $\mathcal{N}_{\text{trunc}}(\mu, \sigma^2, -1, 1)$ distribution. To prove (i) observe that $p_i > \frac{2f(\alpha_i)}{k} > p_{i+1}$.

To prove (ii), divide the $i$'th bin to two sub-bins $B_{i,1}, B_{i,2}$, of lengths $a = \mu - \alpha_{i-1}$ and $b = \alpha_i - \mu$, respectively. Similarly, divide the $i + 1$'th bin to two bins $B_{i+1,1}, B_{i+1,2}$ length $b$ and $a$, respectively. Then from (i)

$$\int_{B_{i,2}} f(x)dx > \int_{B_{i+1,1}} f(x)dx. \tag{8}$$

In addition, observe that

$$\int_{B_{i,1}} f(x)dx = \int_0^a f(\mu + x)dx$$

$$> \int_0^a f(\mu + 2b + x)dx$$

$$= \int_{B_{i+1,2}} f(x)dx. \tag{9}$$

Adding up equation 8 and equation 9, we obtain $p_i > p_{i+1}$. Apart from just being unimodal, we also proved that if $\mu \in (\alpha_{i-1}, \alpha_i)$ then class $i$ is the predicted class by the model.

$\square$

## B    PROOF OF LEMMA 2

*Proof.* Let $-1 = \alpha_0 < \alpha_1 < \ldots < \alpha_k = 1$ be the thresholds. In the proof of Lemma 1 (Appendix A), we demonstrated that for any $i \in \{1, \ldots, k\}$, if $\mu \in (\alpha_{i-1}, \alpha_i)$, then class $i$ is the predicted class by the model.

Given that the two sets of parameters of the truncated normal distribution, $(\mu, \sigma_1^2)$ and $(\mu, \sigma_2^2)$ share the same mean $\mu$ and the thresholds $\alpha_0, \ldots, \alpha_k$ are constant, the predicted class $i$ remains the same for both sets of parameters regardless of the variances $\sigma_1^2$ and $\sigma_2^2$.

$\square$

## C    ADDITIONAL EXPERIMENTS

### C.1    COMPARING UNICORNN AGAINST CORN

Table C.1 describes the results of two experiments of UNICORNN against CORN[2] (Shi et al., 2023) on the Adience and EVA datasets. The technical details of the experiments are the same as the ones that appear in Appendix E. As can be seen, UNICORNN outperforms CORN on both datasets with lower MAE scores..

### C.2    COMPARING UNICORNN AGAINST CORN ON NEW DATASETS

In Table C.2 we provide the MAE results of our method trained on AFAD and Fireman dataset additionally to the results from Shi et al. (2021) for CE-NN, OR-NN, CORAL and CORN. The results for these baselines were borrowed from Shi et al. (2021). We train our model with the same

---

[2]https://github.com/Raschka-research-group/corn-ordinal-neuralnet/blob/main/model-code/simple-scripts/mlp_corn.py

Table 3: UNICORNN vs CORN.

| Method | Dataset | MAE ↓ |
|---|---|---|
| Adience | CORN | .52 ± .07 |
| | UNICORNN | **.46 ± .05** |
| EVA | CORN | .67 ± .02 |
| | UNICORNN | **.57 ± .01** |

backbone as in Shi et al. (2021), Resnet-34 for AFAD dataset, and with two-layer MLP with hidden dimension of size 300 for Fireman dataset. Both datasets are balanced and the train, validation and test splits are the same as in Shi et al. (2021). We train our method 5 times with different random intialization seeds. Our method ourperforms the baselines on both datasets.

Table 4: Test MAE of methods trained on AFAD and Fireman Datasets

| Method | AFAD | Fireman |
|---|---|---|
| CE-NN | 3.28 ± 0.04 | 0.8 ± 0.01 |
| OR-NN | 2.85 ± 0.03 | 0.76 ± 0.01 |
| CORAL | 2.99 ± 0.03 | 0.82 ± 0.01 |
| CORN | 2.81 ± 0.02 | 0.76 ± 0.01 |
| UNICORNN | **2.645 ± 0.02** | **0.755 ± 0.003** |

## C.3 COMPARING UNICORNN AGAINST SIMPLE CONSTRAINT-UNIMODAL BASELINE

We conduct an additional experiment to compare our method with a simple baseline classifier trained using cross-entropy loss and a unimodality constraint. In this baseline, the model predicts a K-dimensional probability vector for each sample, and the unimodality constraint is enforced by minimizing pairwise distances between the predicted probability values. Specifically, the distances are set to be negative for probabilities at indices lower than the target label and positive for indices higher than the target label. Further implementation details are available in the project repository.

The results, shown in Table C.3, indicate that the baseline achieves near-unimodal distributions ( 80% unimodality) on the Fireman dataset but does not guarantee perfect unimodality. Additionally, the Mean Absolute Error (MAE) of this baseline is higher compared to UNICORNN. Notably, when applied to the AFAD dataset with the same training parameters, the baseline fails to produce unimodal predictions, highlighting its limitations in maintaining unimodality across different datasets.

Table 5: Performance Comparison on Balanced Datasets (AFAD and Fireman)

| Method | AFAD (balanced) | | Fireman (balanced) | |
|---|---|---|---|---|
| | MAE ↓ | Unimodality ↑ | MAE ↓ | Unimodality ↑ |
| UNICORNN | 2.64 ± 0.02 | 1.0 ± 0.0 | 0.755 ± 0.003 | 1.0 ± 0.0 |
| Classifier with Unimodality | 2.846 ± 0.04 | 0.11 ± 0.04 | 0.808 ± 0.003 | 0.865 ± 0.05 |

## D DATASETS

Tabel 6 contains information on the benchmark datasets used for our experiments.

**Adience**: During the training the images are resized to (256,256). Additionally, random crop of size 224 and random horizontal flip are applied as augmentations.

**FG-Net**: We partitioned the dataset to 8 classes, corresponding to decades. Augmentations are the same as in the Adience experiment.

**RetinaMNIST** dataset has 5 classes, and we apply random affine, horizontal and vertical flips as augmentations during the training. The size of the images is (28,28) as provided by the dataset contributors. The train/test splits are proved by the contributors and were used as-is.

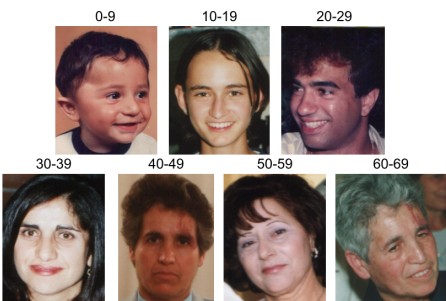

Figure 5: Examples from the FG-Net dataset. Age classes are indicated above each image.

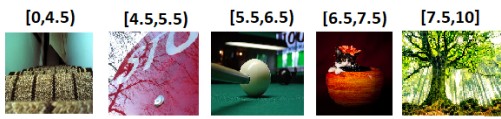

Figure 6: Examples from the EVA dataset. Aesthetics classes are indicated above each image.

**SCUT-FBP5500** dataset contains 5500 face images beautifully ranked from 1 to 5 continuously. we partition the data into 8 classes in accordance with the rank. Augmentations are the same as in the Adience experiment.

**EVA (Explainable Visual Aesthetics)** dataset contains 5101 images aesthetically ranked from 0 to 10 by multiple voters. We calculate the average score for each image and partition the data into 5 classes in accordance with the average score. Augmentations are the same as in the Adience experiment.

**AAF (All-Age-Faces)** dataset is already pre-processed and contains 13,322 face images (mostly Asian), distributed across all ages (from 2 to 80). We partitioned the dataset into 6 classes. Augmentations are the same as in the Adience experiment.

**AFAD (The Asian Face data)**[3] dataset (Niu et al., 2016) contains 165,501 faces in the age range of 15-40 years. No additional preprocessing was applied to this dataset since the faces were already centered. Following Shi et al. (2021), we use a balanced version of the AFAD dataest[4] with 13 age labels in the age range of 18-30 years, resulting in total 60K samples.

**Fireman** [5] dataset is a tabular dataset that contains 40,768 instances, 10 numeric features, and an ordinal response variable with 16 categories. We use a balanced version of this dataset[6] consisting of 2,543 instances per class and 40,688 from the 16 ordinal classes in total.

Figures 5, 6, 7, 8 show examples from the FG-Net, EVA, AAF and SCUT-FBP5500 datasets, respectively.

## E  TECHNICAL DETAILS

Table 7 shows the technical details for the experiments on the real world benchmark datasets reported in this manuscript.

The Adam optimizer was used in all experiments, with the default $\beta = (0.9, 0.999)$. The means and standard deviations reported in table 1 are based on 5 repetitions of each experiment, differing in weights initialization and random train-test splits, except for Adience, for which we repeated the experiment five times, using the same train-test splits as the creators of the dataset[7]. For the

---

[3]https://github.com/afad-dataset/tarball

[4]https://github.com/Raschka-research-group/corn-ordinal-neuralnet/tree/main/datasets/afad

[5]https://github.com/gagolews/ordinal_regression_data

[6]https://github.com/Raschka-research-group/corn-ordinal-neuralnet/tree/main/datasets/firemen

[7]https://github.com/GilLevi/AgeGenderDeepLearning/tree/master/Folds/train_val_txt_files_per_fold

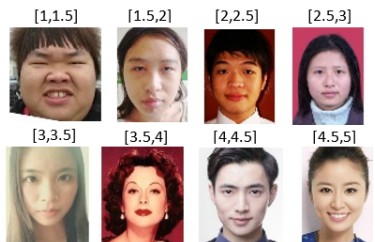

Figure 7: Examples from the AAF dataset. Age classes are indicated above each image.

Figure 8: Examples from the SCUT-FBP5500 dataset. Beauty classes are indicated above each image.

accuracy-preserving calibration phase (Section 4.3), we used early stopping and the number of epochs and learned parameters were changed for each dataset, as shown in Table 8.

# F COMPUTATION DETAILS

We implement our model in Pytorch and run experiments on a Linux server with NVIDIA GeForce GTX 1080 Ti, A100 80GB PCIe GPUs and Intel(R) Core(TM) i7-8700 CPU 3.20GHz CPU.

Table 6: Benchmark datasets characteristics

| Dataset | Task | Train Images | Val Images | Test Images | Classes |
|---|---|---|---|---|---|
| Adience | age estimation | | pre-defined splits | | 8 |
| FG-Net | age estimation | 802 | 100 | 100 | 8 |
| RetinaMNIST | DR classification | 1080 | 120 | 400 | 5 |
| AAF | age estimation | 9058 | 1599 | 2665 | 6 |
| EVA | aesthetics estimation | 3684 | 651 | 766 | 6 |
| SCUT-FBP5500 | facial beauty prediction | 4250 | 350 | 900 | 8 |

Table 7: Technical details of the experiments

| Dataset | Backbone | Epochs | Batch Size | Initial LR | Decay LR After (epochs) | Weight Decay |
|---|---|---|---|---|---|---|
| Adience | ResNet-101 | 100 | 64 | $10^{-4}$ | 40 | $10^{-5}$ |
| FG-Net | ResNet-18 | 100 | 32 | $10^{-4}$ | 40 | $10^{-4}$ |
| RetinaMNIST | ResNet-18 | 100 | 16 | $10^{-4}$ | 80, 90 | $10^{-4}$ |
| AAF | ResNet-18 | 100 | 64 | $10^{-4}$ | - | $10^{-3}$ |
| EVA | ResNet-18 | 145 | 64 | $10^{-4}$ | - | $10^{-5}$ |
| SCUT-FBP5500 | ResNet-18 | 100 | 64 | $10^{-4}$ | - | $10^{-3}$ |

Table 8: Post-hoc calibration technical details

| Dataset | Epochs | Learned Parameters |
|---|---|---|
| Adience | 100 | 250 |
| FG-Net | 100 | 250 |
| RetinaMNIST | 100 | 1000 |
| AAF | 100 | 150 |
| EVA | 100 | 50 |
| SCUT-FBP5500 | 100 | 100 |