# OpenReview forum: "UNICORNN: Unimodal Calibrated Ordinal Regression Neural Network"
_ICLR.cc/2025/Conference — Submitted to ICLR 2025_

### Official Review · Reviewer_jEsc · 2024-10-28

**Soundness:** 3
**Presentation:** 3
**Contribution:** 2
**Rating:** 5
**Confidence:** 3

**Summary:**

This paper introduces UNICORNN, a deep-learning approach designed for ordinal regression tasks where class labels follow a natural order. The authors address three primary challenges in ordinal regression: (1) enforcing unimodal output probability distributions, (2) capturing the ordered relationships among classes effectively, and (3) providing well-calibrated probability estimates. Extensive experiments on six real-world datasets demonstrate that UNICORNN consistently matches or outperforms recent ordinal regression methods, excelling in accuracy, probability calibration, and ensuring output unimodality.

**Strengths:**

1. UNICORNN addresses significant limitations in existing ordinal regression methods by providing an approach that ensures both unimodal and calibrated probabilities, which is critical for applications
2. The paper is well-structured and Explanations are generally clear.
3. The combination of Optimal Transport (OT) loss with accuracy-preserving calibration is original and significant for ensuring both order respect and reliable probability estimates

**Weaknesses:**

1. The proposed method has a two-stage training process, significantly increasing the model's computational complexity. Can the author show the time complexity of training or compare the training time of the proposed method with other methods?
2. The methods used in experiments are too old. The authors should include comparisons with methods from recent years such as [1,2]. Can the author show the results on some large-scale datasets, such as the datasets with more training samples or more classes?
3. More experiments are needed in ABLATION STUDY. It only has two datasets. More experiments on different datasets are required.


[1] Shi X, Cao W, Raschka S. Deep neural networks for rank-consistent ordinal regression based on conditional probabilities[J]. Pattern Analysis and Applications, 2023, 26(3): 941-955.
[2] Li Q, Wang J, Yao Z, et al. Unimodal-concentrated loss: Fully adaptive label distribution learning for ordinal regression[C]//Proceedings of the IEEE/CVF Conference on Computer Vision and Pattern Recognition. 2022: 20513-20522.
[3] Kim D, Chung H, Jang I. Calibration of ordinal regression networks[J]. arXiv preprint arXiv:2410.15658, 2024.

**Questions:**

See weakness

---

> ### Author Response · Authors · 2024-11-21
>
> We thank the reviewer for the time invested into the reviewing process for acknowledging the ability of our method to provide unimodal and calibrated probabilities as well for mentioning our manuscript as well-structured with generally clear explanations. We also appreciate the reviewers' point that the combination of Optimal Transport (OT) loss with accuracy-preserving calibration is original and significant for ensuring both order respect and reliable probability estimates.
> We address the raised issues in detail below.
>
> **(W1) model's computational complexity**
>
> We would like to note that during the calibration phase, we optimize only the parameters of the $\sigma$ head. Notably, the $\sigma$ head has a significantly smaller number of parameters compared to the backbone model, differing by several orders of magnitude. This implies that the increase in the computation complexity is minor since the gradients of the backbone are not computed and no backpropagation is done on the backbone model. For example, in our experiment, we train a model on the AFAD dataset with ResNet-34 architecture. Then we have a backbone with $N=22 \times 10^6$ parameters, and $\sigma$ head with $250 \times 10^3$ parameters, full training complexity time is given by $F + B \cdot (22,250,000)$ where F is the time complexity of the forward pass and B is the time complexity of the backward pass. However, the time complexity for training only the head is $F + B \cdot 250,000$. For larger backbones, the added training time becomes even less significant.
> Additionally, we provide a wall-clock time measurement of full training of our method with and without the calibration phase. As can be seen, the wall-clock calibration time is indeed much less than the one from phase 1.
>
> | Dataset      | Phase 1 Training time HH:mm  | Phase 2 Calibration time HH:mm |
> |--------------|-----------------------------|-----------------------------------|
> | EVA      | 04:03                  | 00:47                    |
> | AAF          | 04:44                  | 00:37                   |
> | FGNET         | 02:14                  | 00:35                   |

---

> ### Author Response · Authors · 2024-11-21
>
> **(W2) Additional up-to-date baselines**
> We thank the reviewer for pointing us to the additional works. Please notice that the used baselines in the manuscript are chosen for their aim to achieve unimodality, some by architecture and others by soft targets, except for POM, which was chosen as it has inspired our work.
> * Regarding the works suggested by the reviewer, we would like to remark that the last work [3] was published concurrently with the ICLR submission dates. Regarding the work in [1], we extend our comparisons by training our method on larger datasets used in [1]: the AFAD images dataset with 60K samples (13 classes) and the Fireman tabular dataset with 40K samples (16 classes). Please notice that AFAD is larger than the datasets we used in our original experiments both in terms of the number of samples and the number of classes. In addition, we evaluate CORN [1] on the Adience and EVA datasets. We provide the results below and add them to Appendix C in the manuscript.
>
> * As for the AFAD and Fireman experiments, We used the same data splits as in [2] for train, validation, and test, and the same backbone architecture. In addition, we didn’t use hyperparameter tuning, and the hyperparameters were chosen to be similar to other image datasets presented in our manuscript (for the AFAD dataset) and the same hyper parameters as in [2] for Fireman. In Table 1 below we present the mean and std values of the MAE metric measured by the baseline methods CE-NN, OR-NN[5], CORAL[4], and CORN[1] together with our model. The results for the baselines were borrowed from [2]. Our model outperforms these methods on both datasets.
>
> In Table 2 below we present the evaluation results of the baseline CORN which were obtained by using the original code release from [1]. We train the model on Adience and EVA datasets and compare against our method.  UNICORNN outperforms all of these baselines in terms of MAE. In addition to the better accuracy performance, UNICORNN also provides well-calibrated probability estimates.
>
>
> Table 1: evaluating our method on AFAD and Fireman datasets.
> | Method       | AFAD (balanced) MAE Mean± Std  ↓ | Fireman (balanced) MAE Mean± Std ↓ |
> |--------------|-----------------------------|-----------------------------------|
> | CE-NN        | 3.28                       ± 0.04                  | 0.8                            ± 0.01                    |
> | OR-NN[5]     | 2.85                        ± 0.03                  | 0.76                           ± 0.01                    |
> | CORAL[4]     | 2.99                        ± 0.03                  | 0.82                           ± 0.01                    |
> | CORN[1]      | 2.81                        ± 0.02                  | 0.76                          ± 0.01                    |
> | UNICORNN (our)          | **2.64                        ± 0.02**                  | **0.755                          ± 0.003**                   |
>
>
> Table 2: evaluating CORN [1] method on Adience and Eva Datasets.
> | Method   | Adience MAE Mean±Std  ↓ | EVA MAE Mean±Std ↓ |
> |----------|-----------------|-------------|
> | CORN[1]  | 0.52      ±         0.07           | 0.67         ±0.02        |
> | UNICORNN (our) | **0.46       ±        0.05**           | **0.57          ± 0.01**        |
>
>
> **(W3) Ablation study**
> We thank the reviewer for the suggestion to extend the ablation study and we provide here more results by extending Table 2 from the manuscript with additional datasets. As the table describes, the calibration phase indeed decreases the ECE, which results in calibrated probability predictions.
>
>
> | Dataset       | Post-hoc Calibration | ECE ↓       |
> |---------------|-----------------------|-------------|
> | Retina MNIST  | yes                  | **.06 ± .01**   |
> |               | no                   | .38 ± .007  |
> | Adience       | yes                  | **.07 ± .03**   |
> |               | no                   | .18 ± .03   |
> | AAF           | yes                  | **.03 ± .01**   |
> |        | no                   | 0.28 ± 0.01 |
> | EVA      | yes                  | **.08 ± .02**   |
> |   | no  | 0.32 ± 0.02 |
>
> **References:**
>
> [1] Shi Xintong, et al "Deep neural networks for rank-consistent ordinal regression based on conditional probabilities." Pattern Analysis and Applications (2023)
>
> [2] Shi, Xintong, Wenzhi Cao, and Sebastian Raschka. "Deep Neural Networks for Rank-Consistent Ordinal Regression Based On Conditional Probabilities." arXiv preprint arXiv:2111.08851 (2021).
>
> [3] Kim D, Chung H, Jang I. Calibration of ordinal regression networks[J]. arXiv preprint arXiv:2410.15658, 2024.
>
> [4] Cao, Wenzhi, et al. "Rank consistent ordinal regression for neural networks with application to age estimation." Pattern Recognition Letters 140 (2020): 325-331.
>
> [5] Z. Niu, et al. Ordinal regression with multiple output CNN for age estimation. In Proceedings of the IEEE Conference on Computer Vision and Pattern Recognition, 2016.

---

### Official Review · Reviewer_fMTJ · 2024-11-03

**Soundness:** 3
**Presentation:** 3
**Contribution:** 2
**Rating:** 5
**Confidence:** 4

**Summary:**

**Summary**:

The paper presents a technique for ordinal regression tasks, focusing on two aspects: (1) constructing uni-modal distribution, which is desirable in many cases; (2) generating calibrated predictive distribution that is comparable to the empirical distribution. For the uni-modal distribution construction, it generates a truncated normal distribution, and constructs evenly spaced thresholds to ensure the unimodality. For the calibration, the model performs retraining using Brier loss to adjust the parameter from the first stage. The authors then demonstrate the performance of the proposed model in the real experiments.

**Strengths:**

**Strengths**:

1. The paper clearly describes the motivations and explanations of the proposed model.
2. A rather simple and interesting approach for ensuring unimodality of the predictive distribution.
3. Retraining procedure for producing calibrated distributions.
4. The experiments show the benefit of the proposed approach on several datasets.

**Weaknesses:**

**Weaknesses**:

1. The description of the proposed training algorithm in Sec. 4.4 is rather simple. I would suggest the authors to put a proper algorithm (using formal notation) for the model
2. The way the authors ensure unimodal distribution can be limiting. Even though the proposed method always generates a unimodal distribution, not every unimodal distribution can be represented by the proposed construction. In fact, with just two degrees of freedom, it is rather limited.
3. The novelty of the proposed models is rather limited, as similar unimodality construction and the calibrated distributions constructions have been proposed before.

**Questions:**

**Questions**:

Please address the weaknesses mentioned above.

---

> ### Author Response · Authors · 2024-11-21
> **Authors' response**
>
> We thank the reviewer for the time invested in the review process and for acknowledging our work in clearly describing the motivations and explanations, as well as proposing an interesting approach to ensuring unimodality of the predictive distribution, supported by experiments that demonstrate the benefits of the proposed method across several datasets.
>  We address the raised issues in detail below.
>
> **(W1) The description of the proposed training algorithm**
>
> We appreciate the reviewer's suggestion and have updated Sec. 4.4 to include two formal algorithms detailing the two phases of training.
>
> **(W2) Two degrees of freedom limits unimodal distribution**
>
> First,  our aim in this work is to propose an improved ability of the model to represent distributions by adding more flexibility with two parameters than previous works, e.g. Beckham et al. [1], but still preserving the overall simplicity of the approach, while providing improved performance and calibrated probability estimates.
>
> We acknowledge that UNICORNN's two degrees of freedom might appear limiting; however, this discussion aligns with the broader trade-off between parametric and non-parametric methods. Where parametric methods learn the parameters of a specific member of a parametric family of distributions, e.g. the $\mathcal{N}(\mu,\sigma)$ family. In contrast, non-parametric methods, such as predicting a histogram directly, make no assumptions about the underlying distribution's functional form. Each approach has its strengths and limitations. While non-parametric methods may offer greater expressiveness, they typically require more data to train effectively. In contrast, parametric methods, such as UNICORNN, require less data. Despite this, our experiments demonstrate that UNICORNN outperforms other non-parametric baselines, such as UnimodalNet and SORD (see Table 1), which highlights its effectiveness.
>
>
>
> **(W3) The novelty of the proposed model**
>
> We would like to present the novelty of our approach concisely:
> Our proposed method, UNICORNN, stands out by seamlessly integrating three critical aspects: (1) enforcing unimodality, (2) maintaining probability calibration, and (3) effectively capturing the ordinal relationships between classes. Unlike prior works that address these aspects either individually or in pairs, UNICORNN is the first approach to achieve all three simultaneously, providing a comprehensive solution to the challenges of ordinal regression. Furthermore, as detailed in Section 3.3, our work highlights a trade-off between OT and calibrated predictions. Specifically, OT may prioritize peaked distributions rather than calibrated ones, demonstrating that these requirements are interdependent. Our novelty is also backed by empirical evidence of the effectiveness of the proposed method.
>
>
> We thank the reviewer for the questions and appreciate any additional comments and feedback.
>
>
> [1] Beckham, C. and Pal, C. (2017). Unimodal probability distributions for deep ordinal classification. In Proceedings of the 34th International Conference on Machine Learning-Volume 70, pages 411–419.

---

### Official Review · Reviewer_rC9M · 2024-11-04

**Soundness:** 2
**Presentation:** 3
**Contribution:** 2
**Rating:** 5
**Confidence:** 5

**Summary:**

In this paper, the authors propose an ordinal regression approach that enforces unimodularity in the output distribution. They use a truncated normal distribution to model the output probabilities.  They provide unimodular proof of the output distribution. They show experimental results to highlight the effectiveness of the proposed approach compared with baseline approaches on various metrics.

**Strengths:**

1. Authors propose a theoretically sound approach for ordinal regression, which can output unimodular outputs.
2. Paper is very well written and easy to read.

**Weaknesses:**

1. Some references are missing. E.g.
(a) Rank consistent ordinal regression for neural networks with application to age estimation. Wenzhi Caoa, Vahid Mirjalilib, Sebastian Raschkaa. Pattern Recognition Letters, (2020), 325-331.
(b) Garg, B. &amp; Manwani, N.. (2020). Robust Deep Ordinal Regression under Label Noise. Proceedings of The 12th Asian Conference on Machine Learning. 129:782-796

2. The baseline approaches used do not cover a variety of algorithms for ordinal regression. Please see the Questions section for suggested baselines.

3. Code is not provided.

**Questions:**

1. Please comment on the rank consistency of the proposed approach. In think, unimodularity of the output distribution and rank consistency of the two contradictory properties. Please comment. It would be good to see the average number of rank violations made by the proposed approach.
2. Please compare with a rank-consistent baseline (e.g. Shi et al., 2023)
3. A simple approach for imposing unimodularity in the output vector is by posing the constraint $P(Y=1)\leq P(Y=2)\leq \ldots \leq P(Y=c)\leq \ldots P(Y=K)$, where $c$ be the target class and $K$ be the total number of classes. Minimizing loss with these constraints can be an alternate approach to achieving unimodularity. It can be a good baseline for the proposed approach. Can you show some results why such an is not good?
4. Please explain the truncated normal distribution properly. What is the support set for this distribution?

---

> ### Author Response · Authors · 2024-11-21
> **Authors' response**
>
> We thank the reviewer for the time invested in the reviewing process and for acknowledging our work as theoretically sound with a well-written and easy-to-read manuscript. We address the raised issues in detail below.
>
> **(W1 + W2) Missing references baselines**
>
> We thank the reviewer for pointing us to these references. We have updated the Related Work section in the manuscript by adding these works [1,2,3](Lines 105-112).
> While the paper consists of experiments on 6 baselines and 6 datasets, we extended our experiments in Appendix C and the tables below based on the references provided by the reviewer (CORAL, CORN [1,3]). These additional experiments further highlight the superiority of UNICORNN over other baselines and across a broader range of datasets.
>
> **(W3) Implementation code**
>
> We will publish our entire code under the paper acceptance to further support research reproducibility. Additionally, we attach our code to this submission with training and evaluation on the Fireman tabular dataset.
>
> **(Q1) Rank consistency and our approach**
>
> First, we would like to emphasize the differences between unimodality and rank consistency. Approaches to ordinal regression can generally be categorized into two types: those that predict the probability mass function (PMF) and those that predict the cumulative distribution function (CDF). In the latter case, rank inconsistencies occur when the predicted CDF is not a non-decreasing function. This results in negative values in the PMF that are derived from the predicted CDF, violating the PMF's definition.
>
> Methods such as CORN and CORAL [1,3] could be determined as those that focus on predicting the CDF and indeed successfully ensure rank consistency. However, the resulting PMF is not guaranteed to be unimodal. Conversely, methods that predict the PMF, such as UNICORNN, inherently avoid rank violations because the CDF derived from their predicted PMF is a non-decreasing function by definition.
>
> This demonstrates that unimodality and rank consistency are not mutually exclusive properties. Certain methods, like UNICORNN, achieve both properties by design.
>
> **(Q2) Rank consistent baseline**
> As suggested by the reviewer, we conducted additional experiments on the AFAD (balanced) image and the tabular Fireman (balanced) datasets studied in [4], we used the same data splits as in [4] for train, validation, and test, and the same backbone architectures (Resnet-34 for AFAD and two-layer MLP with hidden dimension 300 for Fireman). In addition, we didn’t use hyperparameter tuning, and these were chosen to be similar to other image datasets presented in our manuscript for AFAD and similar to the parameters used in [4] for Fireman.
>
> | Method       | AFAD (balanced) MAE Mean± Std  ↓ | Fireman (balanced) MAE Mean± Std ↓ |
> |--------------|-----------------------------|-----------------------------------|
> | CE-NN        | 3.28                       ± 0.04                  | 0.8                            ± 0.01                    |
> | OR-NN[5]     | 2.85                        ± 0.03                  | 0.76                           ± 0.01                    |
> | CORAL[1]     | 2.99                        ± 0.03                  | 0.82                           ± 0.01                    |
> | CORN[3]      | 2.81                        ± 0.02                  | 0.76                          ± 0.01                    |
> | UNICORNN (ours)          | **2.64                        ± 0.02**                  | **0.755                          ± 0.003**                   |
>
>
> Moreover, we conducted another two experiments of UNICRONN against CORN[3], on the Adience and EVA datasets. These experiments’ technical details are the same as the ones we introduced in the manuscript.
>
> | Method       | Adience MAE Mean± Std  ↓ | EVA MAE Mean± Std ↓ |
> |--------------|-----------------------------|-----------------------------------|
> | CORN[3]      | 0.52                        ± 0.07                  | 0.67                          ± 0.02                    |
> | UNICORNN (ours)          | **0.46**                        ± **0.05**                  | **0.57**                          ± **0.01**                  |
>
>
> As can be seen, UNICORNN outperforms all of these baselines in terms of MAE.

---

> ### Author Response · Authors · 2024-11-21
> **Authors' response**
>
> **(Q3) The proposed baseline with unimodality constraint**
>
> First, we assume the reviewer meant to write:
>
> $P(Y=1) \leq P(Y=2) \leq \dots \leq P(Y=c) \geq \dots \geq P(Y=K)$
>
> rather than:
> $P(Y=1) \leq P(Y=2) \leq \dots \leq P(Y=c) \leq \dots \leq P(Y=K)$
>
> as the first term is the correct definition of unimodality.
> Second, incorporating this constraint into the model requires its implementation within the loss function which increases the complexity of tuning the loss function's hyperparameters and imposes an additional challenge on the model to learn unimodality along with accurate predictions.
>
> To the reviewer’s suggestion, we conducted an additional experiment to compare our method with a simple baseline classifier trained using cross-entropy loss and a unimodality constraint. In this baseline, the model predicts a K-dimensional probability vector for each sample, and the unimodality constraint is enforced by minimizing pairwise distances between the predicted probability values. Specifically, the distances are set to be negative for probabilities at indices lower than the target label and positive for indices higher than the target label.
>
> The results, shown in the table below and appendix C.3 in the manuscript, indicate that the baseline achieves near-unimodal distributions (~80\% unimodality) on the Fireman dataset but does not guarantee perfect unimodality. Additionally, the MAE of this baseline is higher compared to UNICORNN. Notably, when applied to the AFAD dataset with the same training parameters, the baseline fails to produce unimodal predictions, highlighting its limitations in maintaining unimodality across different datasets.
>
>
> | Method                                | AFAD (balanced) MAE ↓ | AFAD (balanced) Unimodality ↑ | Fireman (balanced) MAE ↓ | Fireman (balanced) Unimodality ↑ |
> |---------------------------------------|------------------------|-------------------------------|---------------------------|----------------------------------|
> | UNICORNN                              | **2.64 ± 0.02**           | **1.0 ± 0.0**                    | **0.755 ± 0.003**            | **1.0 ± 0.0**                       |
> | Classifier with Unimodality Constraint| 2.8458 ± 0.0358       | 0.1087 ± 0.0376              | 0.808 ± 0.003            | 0.8648 ± 0.0525                 |
>
>
>
>
> **(Q4) Truncated normal distribution and its support.**
>
> Please note that in Lines 309-313, we provide an exact definition of the truncated normal distribution, the CDF of the truncated normal distribution supported on [a,b] is given by:
> $F(x;\mu,\sigma,a,b) = \frac{ \Phi_{\mu,\sigma}(x) - \Phi_{\mu,\sigma}(a) }{ \Phi_{\mu,\sigma}(b) - \Phi_{\mu,\sigma}(a) }$
>
> where $\Phi_{\mu,\sigma}$ is the normal distribution's CDF with mean and std $\mu, \sigma$, respectively. In UNICORNN, we use $a=-1, b=1$.
>
>
> We thank the reviewer for the questions and would be happy to get additional comments and feedback.
>
>
> **References:**
>
> [1] Cao, Wenzhi, et al. "Rank consistent ordinal regression for neural networks with application to age estimation." Pattern Recognition Letters 140 (2020): 325-331.
>
> [2] Garg, B. & Manwani, N. Robust Deep Ordinal Regression under Label Noise. Proceedings of The 12th Asian Conference on Machine Learning. (2020)
>
> [3] Shi Xintong, et al "Deep neural networks for rank-consistent ordinal regression based on conditional probabilities." Pattern Analysis and Applications (2023)
>
> [4] Shi, Xintong, Wenzhi Cao, and Sebastian Raschka. "Deep Neural Networks for Rank-Consistent Ordinal Regression Based On Conditional Probabilities." arXiv preprint arXiv:2111.08851 (2021).
>
> [5] Z. Niu, et al. Ordinal regression with multiple output CNN for age estimation. In Proceedings of the IEEE Conference on Computer Vision and Pattern Recognition, 2016.

---

### Author Response · Authors · 2024-11-21
**General response**

We thank all reviewers for their detailed feedback and thoughtful comments on our work. We are deeply grateful for the recognition of our contributions: the reviewers found our paper to be well-written, and structured, with provided strong motivations and explanations for the proposed model. UNICORNN was recognized as a simple yet effective approach to ensure unimodality and calibrated predictive distributions, addressing key limitations in existing ordinal regression methods. We also value the positive remarks on combining Optimal Transport loss with accuracy-preserving calibration. Our solution to the ordinal regression task was found to be theoretically sound and original, validated through experiments on multiple datasets, which demonstrates its practical benefits. We also have uploaded a revised version of the manuscript, changes appear in color.

In response to the constructive reviews, we have comprehensively addressed the concerns raised to improve the manuscript. Please see our detailed response for each issue below.

Moreover, we conducted new experiments as suggested by the reviewers (**rC9M** and **jEsc**). In the first one, we train our method on new larger datasets: image dataset (AFAD) with 60K samples and tabular dataset (Fireman) with 40K samples. The results indicate that our method outperforms the baselines also on these two new datasets. In an additional experiment, we evaluate one of the suggested recent baselines from [1] on the datasets Adience and EVA we used in the manuscript. In this case, our method outperforms this baseline, even with a wider gap.



[1] Shi Xintong, et al "Deep neural networks for rank-consistent ordinal regression based on conditional probabilities." Pattern Analysis and Applications (2023)

---

### Author Response · Authors · 2024-11-25

Dear reviewers, we appreciate the time and effort you have dedicated to reviewing our paper. As the discussion period is limited, we kindly request that the reviewer evaluate the new information provided in the rebuttal. We are eager to improve our paper and resolve all the concerns raised by the reviewer. If there are any remaining concerns that have not been addressed, we would be happy to provide further explanations.

---

### Meta-Review · Area_Chair_N2Ny · 2024-12-20

**Metareview:**

The paper introduces UNICORNN, an optimal transport-based method for ordinal regression that ensures unimodality and provides calibrated probability estimates while respecting the ordinal nature of the classes. The reviewers found the paper to be well-motivated and well-written, with a theoretically sound method supported by experiments demonstrating its effectiveness across multiple datasets. The main weakness pointed out by the reviewers relates to limited novelty, that the unimodality construction limits the model’s ability to represent diverse distributions. Reviewers also raised concerns about computational overhead, lack of comparison to more recent datasets, and too few datasets used in the ablation study, but these concerns were addressed during the rebuttal and discussion phase. Despite efforts to address reviewer concerns, the paper does not represent a sufficiently substantial advancement over existing methods to warrant acceptance at this time.

**Additional Comments On Reviewer Discussion:**

The main points raised by the reviewers are summarised in the review, and most of these concerns were addressed by the authors.

---

### Decision · Program_Chairs · 2025-01-22

Reject